# Biosynthetic Melanin/Ce6-Based Photothermal and Sonodynamic Therapies Significantly Improved the Anti-Tumor Efficacy

**DOI:** 10.3390/pharmaceutics15082058

**Published:** 2023-07-31

**Authors:** Yuping Yang, Yaling He, Meijun Zhou, Meijun Fu, Xinxin Li, Hongmei Liu, Fei Yan

**Affiliations:** 1The Second School of Clinical Medicine, Southern Medical University, Guangzhou 510515, China; 2Department of Ultrasound, Institute of Ultrasound in Musculoskeletal Sports Medicine, Guangdong Second Provincial General Hospital, Guangzhou 510317, China; 3Department of Ultrasound, Affiliated Hospital of Guangdong Medical University, Zhanjiang 524001, China; 4CAS Key Laboratory of Quantitative Engineering Biology, Shenzhen Institute of Synthetic Biology, Shenzhen Institutes of Advanced Technology, Chinese Academy of Sciences, Shenzhen 518055, China; yl.he@siat.ac.cn (Y.H.); mj.fu@siat.ac.cn (M.F.);

**Keywords:** liposome, biosynthetic melanin nanoparticles, chlorin e6, sonodynamic therapy, photothermal therapy

## Abstract

Photothermal therapy (PTT) and sonodynamic therapy (SDT) are becoming promising therapeutic modalities against various tumors in recent years. However, the single therapeutic modality with SDT or PTT makes it difficult to achieve a satisfactory anti-tumor outcome due to their own inherent limitations, such as poor tissue penetration for the near-infrared (NIR) laser and the limited cytotoxic reactive oxygen species (ROS) generated from conventional sonosensitizers irradiated by ultrasound (US). Here, we successfully biosynthesized melanin with a controllable particle size with genetically engineered bacteria harboring a heat-inducible gene circuit. The biosynthetic melanin with 8 nm size and chlorin e6 (Ce6) was further encapsulated into liposomes and obtained SDT/PTT dual-functional liposomes (designated as MC@Lip). The resulting MC@Lip had an approximately 100 nm particle size, with 74.71% ± 0.54% of encapsulation efficiency for melanin and 94.52% ± 0.78% for Ce6. MC@Lip exhibited efficient ^1^O_2_ production and photothermal conversion capability upon receiving irradiation by US and NIR laser, producing significantly enhanced anti-tumor efficacy in vitro and in vivo. Especially, US and NIR laser irradiation of tumors received with MC@Lip lead to complete tumor regression in all tested tumor-bearing mice, indicating the great advantage of the combined use of SDT and PTT. More importantly, MC@Lip possessed good photoacoustic (PA) and fluorescence dual-modal imaging performance, making it possible to treat tumors under imaging guidance. Our study provides a novel approach to synthesize a melanin nanoparticle with controllable size and develops a promising combined SDT/PTT strategy to treat tumors.

## 1. Introduction

Cancer ranks as a leading cause of mortality worldwide, but efficient therapeutic modalities against cancer have remained a challenge over past decades [1,2]. Although traditional first-line cancer treatment approaches, including chemotherapy, radiotherapy, and surgery, have made great progress in clinical practices, they are still associated with some adverse effects, such as multidrug resistance, immune system impairment, or systemic toxicity [3,4,5]. Recently, the fast development of molecular imaging technology has ushered in new frontiers in the field of anti-cancer therapies. Minimally invasive or noninvasive imaging-guided therapeutic modalities, such as microwave therapy, phototherapy, and acoustic therapy, are attracting researchers’ attention and showing promising potential for cancer treatment due to their precisely spatiotemporal controllability, special tumor-killing activity, and elevated therapeutic outcomes [6,7,8].

Recently, sonodynamic therapy (SDT), a noninvasive treatment modality through combining sonosensitizers with low-intensity ultrasound (US), is rapidly developing to be a promising approach for oncotherapy. Thanks to the merits of deep tissue-penetrating capability (>10 cm), low cost, and non-invasiveness, SDT has been demonstrated to be more effective than photodynamic therapy (PDT), which suffers from the main limitation in the low-light penetration of tissue depth [9,10,11]. The predominant mechanism of SDT is considered to be the production of cytotoxic reactive oxygen species (ROS) from sonosensitizers irradiated by US, such as cytotoxic singlet oxygen (^1^O_2_), leading to damage to intracellular DNA and apoptosis of cancer cells [12,13]. So far, numerous sonosensitizers, including organic and inorganic materials, have been developed [14]. Among them, chlorin e6 (Ce6), an FDA-approved organic sensitizer of porphyrin derivatives [15], shows some unique advantages over other sonosensitizers, including effective activation by US and fluorescence imaging capability, which makes it possible to realize imaging-guided SDT for precise tumor therapy [16,17,18,19]. Unfortunately, SDT still has some problems. For example, the hypoxic tumor microenvironment (TME) may restrict the therapeutic efficacy of SDT due to the continuous O_2_ consumption during SDT [20,21]. In addition, the short diffusion distance of ROS also makes it difficult to kill large-scale tumor cells [22,23]. All these disadvantages mentioned above show that there is still a challenge for the optimal therapeutic outcome by only SDT.

Unlike SDT, photothermal therapy (PTT) is another promising oxygen-independent anti-tumor strategy, which employs photothermal agents to transduce light energy into hyperthermia to kill tumors [24,25]. The application of PTT to tumor treatment is highly effective; however, the limited tissue penetration of PTT usually results in incomplete ablation for some large tumors, which greatly decreases its anti-tumor efficacy [26]. Thus, SDT or PTT, as a single therapeutic modality, cannot achieve a satisfactory outcome. Previous studies have reported that hyperthermia-induced enhancement of blood perfusion during PTT could alleviate hypoxia and promote SDT-mediated ROS generation [27,28]. Documents have also demonstrated that the combined use of PTT and SDT could overcome the inherent limitations of the single PTT or SDT modality and achieve significantly improved anti-tumor effects [29,30]. To date, many photothermal agents with strong near-infrared (NIR) absorption, such as Au, Pt, and Pd metal nanoparticles (NPs), have been developed for PTT treatment of tumors [31,32,33]. Nevertheless, most of them face the dilemma of potential tissue toxicity and low photothermal conversion efficiency.

Melanin is a natural polyphenol substance existing in biological tissues, including the hair and skin of mammals, the ink of cuttlefish, black sesame, bacteria, etc. [34,35,36,37,38]. Also, melanin is an endogenous chromophore with photoacoustic (PA) imaging capability, exhibiting high sensitivity and spatial resolution in in vivo deep-seated tissues [39]. Generally, there are two main ways to obtain melanin NPs. One is chemical synthesis, and the other is biological extraction. The former is environmentally unfriendly and has poor biocompatibility, and the latter is procedurally complex, low efficiency, and expensive. Herein, we explored a biosynthetic route to obtain melanin NPs via genetically engineered bacteria and fabricated the SDT/PTT dual-functional liposomes (designated as MC@Lip) by encapsulating the biosynthetic melanin NPs into a hydrophilic core and Ce6 into the hydrophobic lipid shell. The genetic circuit, which harbors a tyrosinase gene named Tyr1 under the control of a TlpA promoter inhibited by the temperature-sensitive repressor protein (TlpA36), was introduced into E.coli. The genetically engineered bacteria can express tyrosinase in a temperature-induced manner, catalyzing Tyrosine substrate to convert into melanin. The relatively slow biosynthetic procedure makes it possible to fabricate different particle sizes of melanin NPs by controlling the incubation time of bacteria with tyrosine substrate. The biosynthetic melanin NPs with 8 nm particle size and Ce6 sonosensitizer were further encapsulated into liposomes via the filming-rehydration method and obtained the MC@Lip. After these resulting MC@Lip were further intravenously administrated into the tumor-bearing mice, they would target the tumor site by the enhanced permeability and retention (EPR) effect. Upon the MC@Lip becoming enriched in the tumor, they would be received with acoustic and laser irradiation, activating the combined treatments of SDT and PTT and greatly improving the anti-tumor efficacy (Figure 1).

## 2. Materials and Methods

### 2.1. Materials

1,2-distearoyl-sn-glycero-3-phosphoethanolamine-N-[methoxy(polyethylene glycol)-2000] (DSPE-PEG2000), 1,2-dipalmitoyl-sn-glycero-3-phosphatidylcholine (DPPC), and cholesterol were bought from Avanti Polar Lipids Inc (Alabaster, AL, USA). Calcein-AM, propidium iodide, and the CCK-8 (Cell Counting Kit-8) assay were purchased from Dojindo Laboratories (Kumamoto, Japan). 2,7-Dichloro-fluorescein diacetate (DCFH-DA) was purchased from Beyotime Biotechnology (Shanghai, China). Chlorin e6 (Ce6) and Singlet Oxygen Sensor Green (SOSG) were bought from Meilun Biotechnology Co., Ltd. (Dalian, China). 2,2,6,6-Tetramethylpiperidine (TEMP), 4, 6-diamidino-2-phenylindole (DAPI), and L-tyrosine were purchased from Sigma Aldrich (St. Louis, MO, USA). 1.3-diphenylisobenzofuran (DPBF) was bought from Solarbio Biotechnology Co., Ltd. (Beijing, China). Copper (II) Sulphate Pentahydrate was purchased from Titan Scientific Co., Ltd. (Shanghai, China). The sodium dodecyl sulfate-polyacrylamide gel electrophoresis (SDS-PAGE) Gel Fast Preparation Kit was bought from Epizyme Biomedical Technology Co., Ltd. (Shanghai, China).

### 2.2. Synthesis of Biosynthetic Melanin

The tyrosinase gene named Tyr1 from Bacillus megaterium was synthesized by Kingsley Genetics and cloned into the pJC vector under the control of the pTlpA promoter by a Seamless Cloning Kit (D7010M, Beyotime Biotechnology, Shanghai, China), followed by transformation into *E. coli* BL21. Tyrosinase expression was induced at OD 0.6 in a shaking incubator at 25 °C, 30 °C, or 37 °C for different times, followed by analysis with SDS-PAGE. The gel was visualized by staining with Coomassie brilliant blue solution after electrophoresis. Melanin was subsequently synthesized by supplementing it with 0.04 mg mL^−1^ CuSO_4_, as well as 0.8 mg mL^−1^ L-tyrosine, when the genetically engineered bacteria at OD 0.6 in a shaking incubator at 37 °C. The bacterial cells were firstly removed by centrifugation (7000 rpm, 8 min), and the black supernatant containing melanin nanoparticles was further passed through a Millipore 0.22 μm polyethersulfone membrane to remove cell debris. Afterwards, the black supernatant was washed with deionizing water several times by using a 10 K ultrafiltration membrane with centrifugation (7000 rpm, 5 min), and the final melanin solution was harvested and lyophilized for further use.

### 2.3. Characterization of Biosynthetic Melanin

The morphological characterization of melanin nanoparticles was examined by TEM at an acceleration voltage of 100 kV (JEM-1200EX). Dynamic light scattering (DLS) measurement (Zeta Sizer, Malvern, PA, USA) was applied to estimate the size distribution of melanin nanoparticles. UV-Vis-NIR absorbance spectra of biosynthetic melanin were determined at 25, 50, 100, 200, 400, or 800 μg mL^−1^ concentrations. The linear relationship between the concentration of biosynthetic melanin and the absorbance in the UV-Vis-NIR region was plotted at the wavelength 808 nm. Fourier-transform infrared spectroscopy (FTIR) was applied to identify the functional groups of biosynthetic melanin. Briefly, 1 mg melanin powder was mixed with 160 mg anhydrous potassium bromide (KBr; Specac, Orpington, UK), which was compressed into a translucent pellet with a hydraulic press (HY-12). The transmitted FTIR measurements (INVENIO R+RAMII, Bruker, Karlsruhe, Germany) were performed in an air background.

### 2.4. Fabrication of MC@Lip

Biosynthetic melanin and Ce6-loaded liposomes (termed MC@Lip) were fabricated through the thin-film hydration method. Briefly, 0.8 mg Ce6 in 1 mL methanol was added into the 1 mL CHCl_3_-containing lipid mixture (DPPC, cholesterol, and DSPE-PEG-2000, weight ratio at 3:1:1). Then, this solution was transferred into a round-bottom tube. The solvent was removed with nitrogen gas flow to form a uniform thin film, followed by vacuum drying for 2 h. Afterwards, the film was hydrated in 2 mL of melanin solution (8 mg mL^−1^) at 37 °C. The MC@Lip was harvested via mini-extruders under an extrusion method with the 200 nm and 100 nm membranes separately. The homogenized liposomes were further purified via a centrifugal filter (MWCO = 50 kDa) at 4000× *g* for 10 min. After that, the MC@Lip was re-dispersed with deionized water, and the final liposomes were attained and stored at 4 °C in the dark for further use. The unloaded, Ce6-loaded, or melanin-loaded liposomes were also prepared by this similar process.

### 2.5. Characterization of MC@Lip

The structure and morphology of MC@Lip were examined by transmission electron microscopy (TEM, JEM-1200EX, Tokyo, Japan). The photo absorption spectra and fluorescence spectra of Ce6 and biosynthetic melanin were recorded using a microplate spectrophotometer (Multiskan GO, Thermo Scientific, Waltham, MA, USA) and multi-mode microplate reader (Bio Tek Synergy4, Winooski, VT, USA), respectively. The hydrodynamic particle size distribution and zeta potential were measured by using a Malvern Zetasizer Nano S90 instrument (Malvern, Worcestershire, UK). The particle sizes of MC@Lip dispersed in phosphate-buffered saline (PBS), fetal bovine serum (FBS), or water were recorded after 0, 1, 3, 5, or 7 days. The standard concentration curves of Ce6 and biosynthetic melanin solutions at a wavelength of 404 nm and 808 nm were measured to evaluate the amount of Ce6 and biosynthetic melanin encapsulated into MC@Lip. The supernatant was collected for absorbance tests at 404 nm and 808 nm to determine the mass of free Ce6 and melanin, respectively. The encapsulation efficiency of Ce6 and melanin were calculated by equation as follows: encapsulation efficiency (%) = (Weight of Total Ce6 or melanin inputs-Weight of Free Ce6 or melanin)/Total Weight of Ce6 or melanin inputs × 100%.

### 2.6. In Vitro ^1^O_2_ Generation from MC@Lip by US Irradiation

For the quantitative analysis of ^1^O_2_ generation, MC@Lip was suspended in PBS (Ce6 at the concentration of 12 μg mL^−1^), and DPBF (50 µM) was added. Then, the mixture was irradiated by US (1.0 MHz, 1.7 W cm^−2^, 50% duty cycle) for 5 min in the dark. Finally, the absorbance intensity at the wavelength of 398 nm in each group was measured via a multi-mode microplate reader. The DPBF consumption was calculated by the following equation: the consumption of DPBF (%) = (the before irradiation absorbance of DPBF—the post irradiation absorbance of DPBF)/the before irradiation absorbance of DPBF × 100%. Additionally, the MC@Lip solution was added into the SOSG (5 µM), and then the mixture was treated under US irradiation (1.0 MHz, 1.7 W cm^−2^, 50% duty cycle) at various time durations (0 s, 30 s, 60 s, 90 s, or 120 s) to evaluate the ^1^O_2_ generation. A multifunctional microplate reader (λex =488 nm, λem = 525 nm) was used to detect the fluorescence (FL) intensity of ROS generation. The ^1^O_2_ generation from MC@Lip under US irradiation was also detected by TEMP. Typically, MC@Lip (Ce6 at the concentration of 12 μg mL^−1^) with the TEMP (50 mM) was irradiated with US (1.0 MHz, 1.7 W cm^−2^, 50% duty cycle) for 1 min. After that, the A300-10/12 electron spin resonance (ESR) spectrometer (Bruker, Karlsruhe, Germany) was applied to immediately detect the ^1^O_2_ generation. The melanin + TEMP as the control group was also tested.

### 2.7. In Vitro Photothermal Effect of MC@Lip

The photothermal effect of MC@Lip in vitro was evaluated after 808 nm NIR laser irradiation of MC@Lip with different concentrations of melanin. The distance between the laser and sample was fixed at 10 cm. Additionally, the 808 nm NIR laser with various power densities was applied for laser irradiation. During this test, water, blank lip, free Ce6, and melanin were set as controls. An infrared thermal imaging camera (FLIR E4) with an accuracy of 0.1 °C was used to record the temperature variations and infrared radiation (IR) thermal images.

### 2.8. Cytotoxicity of MC@Lip

Dulbecco’s Modified Eagle Medium (DMEM) added with 1% penicillin/streptomycin and 10% FBS was used to culture the murine breast cancer 4T1 cell line and normal bEnd.3 cell line at 37 °C under a 5% CO_2_ atmosphere. Cells in the exponential growth phase were further applied for the experiments. To evaluate the cytotoxicity of MC@Lip, 4T1 and bEnd.3 cells were seeded into the 96-well plate with a density of 1 × 10^4^ per well overnight. After cell adhesion, the cell medium was replaced with a new medium containing MC@Lip at various melanin concentrations (0, 25, 50, 100, 200, 400, and 600 μg mL^−1^). After 24 h co-incubation, the standard CCK-8 assay based on the wavelength of 450 nm absorbance via a multi-mode microplate reader was used to test cell viabilities.

### 2.9. Cell Uptake of MC@Lip and ^1^O_2_ Generation

4T1 cells were incubated with MC@Lip (melanin at the concentration of 200 μg mL^−1^) or melanin solution at 200 μg mL^−1^ for 4 h, and then cells were washed twice and imaged by an inverted fluorescent microscope (Thermo Fisher Invitrogen EVOSM5000, Taunton, MA, USA). Intracellular ^1^O_2_ production upon exposure to US irradiation was detected by a ROS assay kit (DCFH-DA). 4T1 cells were seeded into the 96-well culture plates with a density of 1 × 10^4^ cells per well and divided into four groups: the group without any treatment (Control), only US irradiation group (US), MC@Lip group (MC@Lip), and MC@Lip under US irradiation group (MC@Lip + US). After incubation with MC@Lip (Ce6 at the concentration 12 μg mL^−1^ in DMEM) at 37 °C for 4 h, DCFH-DA was used to replace the culture media, and the cells were exposed to (1.0 MHz, 1.7 W cm^−2^, 50% duty cycle) for 1 min. Next, 30 min later, the cells were gently washed with PBS three times and then imaged by inverted fluorescent microscope to evaluate the production of ^1^O_2_.

### 2.10. In Vitro Tumor Cell-Killing Experiments

4T1 cells were seeded in 96-well culture plates with a density of 1 × 10^4^ cells per well and then divided into six groups: the group without any treatment (Control), MC@Lip group (MC@Lip), US combined with NIR irradiation group (US + NIR), MC@Lip combined with US irradiation group (MC@Lip + US), MC@Lip combined with NIR irradiation group (MC@Lip + NIR), and MC@Lip combined with US and NIR irradiation group (MC@Lip + US + NIR). For the control group, cells were cultured with serum-free culture medium. For the MC@Lip group, cells were cultured with MC@Lip solution (melanin concentration of 200 μg mL^−1^) for 4 h without any irradiation. For the US + NIR group, cells were treated by US irradiation (1.0 MHz, 1.7 W cm^−2^, 50% duty cycle, 1 min) and to NIR irradiation (808 nm and 1 W cm^−2^, 5 min) without MC@Lip. For the MC@Lip + US group, cells were cultured with MC@Lip solution for 4 h with only US irradiation (1.0 MHz, 1.7 W cm^−2^, 50% duty cycle, 1 min). For the MC@Lip + NIR group, cells were cultured with MC@Lip solution for 4 h with only NIR irradiation (808 nm and 1 W cm^−2^, 5 min). For the MC@Lip + US + NIR group, cells were cultured with MC@Lip solution for 4 h with irradiation of US (1.0 MHz, 1.7 W cm^−2^, 50% duty cycle, 1 min) and NIR (808 nm, 1 W cm^−2^, 5 min). After 7–8 h, the standard CCK-8 assay was applied to test cell viabilities. Meanwhile, after the above treatments, cells of different groups were also stained with Calcein-AM and PI solutions, showing living cells fluoresced green and dead cells fluoresced red, respectively, in the inverted fluorescent microscope.

### 2.11. In Vivo Fluorescence and PA Imaging

All animals (female BALB/c mice: 16~18 g, 4–6 weeks) were bought from Zhuhai Bai shi tong Biotechnology Co. Ltd. All in vivo experiments were carried out under the approved protocols of the Shenzhen Institutes of Advanced Technology and the Animal Care and Use Committee of the Chinese Academy of Sciences. As for the establishment of the subcutaneous xenograft mice tumor model, 4T1 cells with a density of 1 × 10^6^ were subcutaneously administrated into the right flank of each female BALB/c mouse. After seven days, the tumors volumes were allowed to reach ~80 mm^3^ before the experiments. For the PA imaging of MC@Lip in vivo, 4T1 tumor-bearing mice (*n* = 3) were intravenously administrated with MC@Lip dispersion at the melanin dose of 60 mg kg^−1^ (200 μL). PA imaging at tumors was acquired at different time points (0 h, 2 h, 4 h, 8 h, 12 h, and 24 h) after injection by using a Vevo LAZR X system. Both PA and ultrasound (UA) signals were acquired with the laser wavelength set at 700 nm. Moreover, quantitative analysis of PA signals at the tumors was performed correspondingly. For in vivo fluorescence imaging of MC@Lip, 4T1 tumor-bearing mice (*n* = 3) were intravenously injected with MC@Lip dispersion at the Ce6 dose of 3.8 mg kg^−1^ (200 μL). Fluorescence images in the tumor regions were recorded at different times post injection (0 h, 2 h, 4 h, 8 h, 12 h, and 24 h) by using a fluorescence system (SPECTRUM, Caliper, Hopkinton, MA, USA). After 24 h, the major organs and tumor site of mice were dissected to obtain fluorescence images ex vivo. Moreover, the relative fluorescence intensities of the major organs and tumor site were measured correspondingly. In addition, tumors were dissected for ultrathin sections and stained with DAPI for fluorescence microscopy detection of Ce6.

### 2.12. In Vivo Anti-Tumor Efficacy of MC@Lip

The 4T1 tumor-bearing mice were divided into six groups at random (*n* = 5), including the control group, MC@Lip group, US + NIR group, MC@Lip + US group, MC@Lip + NIR group, and MC@Lip + US + NIR group. Mice in the control and US + NIR groups were intravenously injected with PBS. The mice in the MC@Lip, MC@Lip + US, MC@Lip + NIR, and MC@Lip + US + NIR groups were administered with 200 μL MC@Lip dispersion at the Ce6 dose of 3.8 mg kg^−1^ and melanin dose of 60 mg kg^−1^. As for the MC@Lip + US group, US irradiation (1.0 MHz, 1.7 W cm^−2^, 50% duty cycle, 5 min) was conducted at 4 h post injection. As for the MC@Lip + NIR group, NIR irradiation (808 nm, 1.0 W cm^−2^, 10 min) was conducted at 4 h post injection. In the US and NIR irradiation groups (US + NIR and MC@Lip + US + NIR), US irradiation (1.0 MHz, 1.7 W cm^−2^, 50% duty cycle, 5 min) was conducted and then exposure to NIR irradiation (808 nm, 1.0 W cm^−2^, 10 min) at 4 h post injection. The temperature changes and IR thermal images of the tumor region were recorded by FLIR E4. The body weight and tumor size of mice were recorded and calculated every other day after treatments. The tumor volume (V) was measured by the following formula: V = length × width^2^/2. The changes of tumor volume and body weight were normalized using the relative tumor volumes named V/V_0_ (V_0_ represented the initial tumor volume before the treatment) and relative body weight W/W_0_ (W_0_ represented the initial body weight before the treatment), respectively, for 20 days. The survival of mice was monitored for 60 days for evaluation of the anti-tumor efficacy. When the tumor volume of mice reached more than 2000 mm^3^, mice were then sacrificed, and the tumor sites were sectioned into slices for staining with hematoxylin and eosin (H&E), Ki-67, and the TdT-mediated dUTP Nick-End Labeling (TUNEL) assay. To sum up, the subcutaneous xenograft mice tumor model was initially established by subcutaneously administrating 4T1 cells into the right flank of female BALB/c mouse. Seven days later, after 4 h post injection of MC@Lip, the 4T1 tumor-bearing mice were treated with 5 min ultrasound firstly and then 10 min NIR irradiation to achieve MC@Lip-mediated SDT and PTT.

### 2.13. Blood and Histological Analysis

To analyze the in vivo biosafety, healthy BALB/c mice (*n* = 3) were intravenously administrated with 200 μL of MC@Lip dispersion at the Ce6 dose of 3.8 mg kg^−1^ and melanin dose of 60 mg kg^−1^. The other three healthy mice were employed as the control group without any treatment. After 14 days, they were sacrificed. The blood samples were used for biochemistry analysis and major organs, including the heart, liver, lung, spleen, and kidney, were harvested for H&E staining.

### 2.14. Statistical Analysis

GraphPad Prism 6 software was used to conduct statistical analyses. The data are shown as the mean value ± standard deviation (SD). Significance was evaluated via Student’s *t*-test or one-way ANOVA with Tukey’s post hoc test. The asterisks (* *p* < 0.05, ** *p* < 0.01, *** *p* < 0.001) represent statistical significance, while n.s. is considered no significance.

## 3. Results

### 3.1. The Preparation and Characterization of Biosynthetic Melanin

The tyr1 gene with the open reading frame from B. megaterium was synthesized (Appendix A). From Figure 2A, we can see that there was no protein band of tyrosinase in the gel of SDS-PAGE after these genetically engineered bacteria were incubated at 25 °C or 30 °C for 8 h. By contrast, the protein band of tyrosinase at 28 kDa could be observed in the gel of SDS-PAGE after these bacteria were exposed to 37 °C for 2 h and gradually became more obvious after 4 h or 8 h. Moreover, after adding L-tyrosine and CuSO_4_ to the bacteria cultivated at the different temperatures of 25 °C, 30 °C, or 37 °C, we found that the suspension color of 37 °C cultivated bacteria become black after 12 h, while the 25 °C and 30 °C cultivated bacterium suspension did not (Figure 2B,C). The particle size of biosynthetic melanin by transmission electron microscopy (TEM) increased from 8 to 35 nm when the incubation time with L-tyrosine was prolonged from 12 h to 24 h (Figure 2D), which was further confirmed by DLS assay (Figure 2E). In addition, the photo absorption of biosynthetic melanin nanoparticles at various concentrations was broad from the ultraviolet region to the NIR region and had a good linear relationship (Figure 2F,G). The FTIR spectra revealed the major functional groups of biosynthetic melanin (Figure 2H).

### 3.2. The Preparation and Characterization of MC@Lip

Next, we encapsulated the biosynthetic melanin with 8 nm particle size and Ce6 into the liposome and obtained the melanin and Ce6-loaded liposomes (MC@Lip) by thin-film hydration (Figure 3A). As the control groups, we also prepared three liposomal nanoparticles, including blank liposomes (Lip), Ce6-loaded liposomes (Ce6@Lip), and melanin-loaded liposomes (Mel@Lip). The morphology of the resulting MC@Lip by TEM displayed a spherical shape, with an average diameter of ~100 nm (Figure 3B). The small biosynthetic melanin nanoparticles loaded into MC@Lip could clearly be observed, indicating the successful encapsulation of melanin (Figure 3C). The 500 nm particle size would be generated when 35 nm biosynthetic melanin nanoparticles were used (Figure 3D). Like free melanin, MC@Lip also exhibited strong photo absorption from the UV to NIR region, while Ce6 displayed the absorption peak at 401 nm, and Lip revealed almost no photo absorption. Compared to the Ce6 or melanin, the MC@Lip remained a strong photo absorption of melanin, and a new characteristic absorption centered at 403 nm corresponded well to that of Ce6, confirming the successful encapsulation of Ce6 and melanin into the MC@Lip (Figure 3E). The encapsulation efficiencies of melanin and Ce6 in MC@Lip were 74.71% ± 0.54% and 94.52% ± 0.78%, respectively.

Also, both the MC@Lip and Ce6@Lip displayed strong fluorescence under the emission peaks at the respective 680 and 662 nm (Figure 3F). The photo-absorption spectra of the Ce6 and MC@Lip solutions are presented in Appendix A. Moreover, the linear relationship between the optical density and concentration was desirable (Appendix A). The mean hydrodynamic diameters of Lip, Ce6@Lip, Mel@Lip, and MC@Lip were 81.52 ± 2.88, 91.12 ± 1.93, 84.31 ± 4.93, and 97.27 ± 7.85 nm, respectively (Figure 3G), revealing no significant size differences in these formulations. The zeta potential changes from −39.80 ± 2.10 mV for Mel@Lip and −32.50 ± 0.40 mV for Ce6@Lip to −42.37 ± 0.90 mV for MC@Lip (Figure 3H). After being stored at 4 °C for seven days, MC@Lip in water, phosphate-buffered saline (PBS), and 10% fatal bovine serum still displayed a favorable stability, as displayed by the negligible size change and low polydispersity index (PDI) (Figure 3I and Appendix A).

### 3.3. Sonodynamic and Photothermal Effects of MC@Lip

Firstly, MC@Lip was exposed to the US irradiation at 1.7 W cm^−^^2^ for 5 min, and then the 1,3-diphenylisobenzofuran (DPBF) consumption was used to analyze US-triggered generation of ^1^O_2_. From Figure 4A, we can see that there was about 47% of DPBF to be oxidized in the US-irradiated MC@Lip, significantly higher than that of only the US group and MC@Lip without US group. The typical SOSG sensor was also employed to determine US-mediated ^1^O_2_ production for MC@Lip irradiated for different durations, revealing time-dependent ^1^O_2_ production (Figure 4B). Moreover, the electron spin resonance (ESR) analysis by the 2,2,6,6-tet-ramethylpiperidine (TEMP) probe revealed the strong ^1^O_2_-induced characteristic signal peaks for MC@Lip exposed to US irradiation, while there was no noticeable peak in the melanin with US group (Figure 4C).

Secondly, from Figure 4D and Appendix A, we found that the temperature elevation (ΔT) achieved 23.4 °C in the MC@Lip + Laser group, compared to the free melanin group (ΔT = 17.5 °C) and Lip group (almost no temperature elevation). To investigate the favorable PTT concentrations of MC@Lip, MC@Lip containing various melanin doses (50, 100, 200 μg mL^−^^1^) was exposed to 808 nm NIR irradiation at the fixed power density of 1.00 W cm^−^^2^. Figure 4E and Appendix A show the melanin concentration-dependent photothermal performance of MC@Lip, achieving the most remarkable ΔT (23.4 °C) for the MC@Lip with 200 μg mL^−^^1^ melanin after 5 min NIR irradiation, while the irradiation of deionized water exhibited almost no variation at ΔT (1.2 °C). Similarly, the temperature increases of MC@Lip at various 808 nm laser power densities (0.75, 1.00, 1.25, 1.50 W cm^−^^2^) were also recorded with the fixed melanin concentration of 200 μg mL^−^^1^, showing significant temperature elevation, along with an increase in NIR irradiation power (Figure 4F and Appendix A). The photothermal stability of MC@Lip was also assessed through repeated exposure to the NIR laser for five laser on/off cycles (808 nm, 1.00 W cm^−^^2^, 5 min each). Figure 4G clearly shows that MC@Lip could maintain the same level of temperature elevation, suggesting excellent photostability for MC@Lip. As shown in Figure 4H, the temperature equilibrium time during the laser on/off process reached to 5 min. The photothermal conversion efficiency (η) of MC@Lip was measured with the fitting curve of time vs. −Ln(θ) derived from the cooling stage (Figure 4I).

### 3.4. In Vitro SDT and PTT Capability of MC@Lip

We firstly examined the cytotoxicity of MC@Lip to 4T1 breast cancer cells and bEnd.3 normal cells at various concentrations with a standard CCK-8 assay. After incubating for 24 h, MC@Lip exhibited a negligible toxicity to both 4T1 cells and bEnd.3 cells at all tested concentrations (0–600 μg mL^−1^) (Figure 5A). After that, 4T1 cells were used to incubate with MC@Lip for 4 h to evaluate their cell uptake efficiency. The MC@Lip- but not melanin-incubated cells displayed the strong red fluorescence of Ce6 in cytoplasm (Figure 5B). Next, the intracellular ROS generation of MC@Lip under US stimulation was assessed by using 2,7-Dichloro-fluorescein diacetate (DCFH-DA) as a ROS molecular probe. From Figure 5C, we can see that strong green fluorescence appeared in the MC@Lip + US group, while negligible fluorescence signals were shown in the only US and MC@Lip groups.

To investigate the SDT and PTT efficiency of MC@Lip, 4T1 cells were incubated with or without MC@Lip and then received with US or/and laser irradiation. From Figure 5D, we can see that the untreated control and MC@Lip groups remained high, with more than 90% cell viability, just like the US + NIR group. By contrast, the MC@Lip + US and MC@Lip + NIR groups exhibited strong cytotoxicity to 4T1 cells. Moreover, the MC@Lip + US + NIR group displayed the strongest cytotoxicity, with less than 10% viable 4T1 cells. The Calcein AM/PI staining assay further evaluated the tumor cell-killing effect of MC@Lip-mediated SDT and PTT. Almost all cells treated with MC@Lip + US+ NIR displayed strong red fluorescence, confirming their excellent cytotoxicity to 4T1 cells (Figure 5E).

### 3.5. In Vivo Photoacoustic and Fluorescence Imaging of MC@Lip

To demonstrate the PA imaging capability of MC@Lip, 4T1 tumor-bearing mice were intravenously administrated with MC@Lip, followed by in vivo PA imaging at tumor sites at different time points. Figure 6A,B clearly reveal that the PA signals of the tumor gradually increased with time and reached a plateau at approximately 4 h after post injection, in consistence with the previous melanin-related report [34,35]. Given that Ce6 has fluorescence imaging capability, we further detected the fluorescence imaging performance of MC@Lip in the tumor. As showed in Figure 6C, the strongest fluorescence signal of the tumor could be observed at 4 h post injection due to the EPR effect (Figure 6C,D), in consistence with the in vivo PA imaging of MC@Lip. Additionally, the tumors and major organs were obtained at 24 h post injection to investigate their ex vivo fluorescence imaging and the biodistribution of MC@Lip. Interestingly, Figure 6E,F reveal substantial MC@Lip was retained in the tumor site, thanks to its high stability and long blood circulation duration. Histological analysis of tumor sections showed that lots of red fluorescence signals from MC@Lip could be observed in the tumor received with MC@Lip systemic administration. By contrast, no red fluorescence signals could be found in tumor sections from the PBS control group (Figure 6G).

### 3.6. In Vivo Photothermal and Sonodynamic Treatment of Tumor

Figure 7A presents the schematic diagram of the in vivo experimental procedure. In detail, the subcutaneous xenograft mice tumor model was initially established by subcutaneously administrating 4T1 cells into the right flank of female BALB/c mouse. Seven days later, after 4 h post injection of MC@Lip, the 4T1 tumor-bearing mice were treated with 5 min ultrasound firstly and then 10 min NIR irradiation to achieve MC@Lip-mediated SDT and PTT. The survival of mice was monitored subsequently for evaluation of the anti-tumor efficacy. Before treatment of the tumor, we firstly optimized the in vivo NIR irradiation time to ensure the photothermal performance of MC@Lip in the tumor-bearing mice. MC@Lip or PBS was intravenously administrated into the tumor-bearing mice and followed by NIR (808 nm, 1.00 W cm^−2^) irradiation of the tumors at 4 h post injection. The tumor temperature change was recorded by an infrared (IR) thermal camera. From Figure 7B,C, we can see that the surface temperature of the tumor went up from 29.1 ± 0.5 °C to 52.7 ± 1.3 °C along with the laser irradiation time, achieving a temperature elevation platform until 10 min (ΔT over 20 °C). By comparison, the surface temperature of the tumor in PBS-injected mice only elevated to around 4.8 ± 0.5 °C after 10 min of laser irradiation. Then, the 4T1 tumor-bearing mice were randomly divided into six groups, including: Control group, MC@Lip group, US + NIR group, MC@Lip + US group, MC@Lip + NIR group, and MC@Lip + US + NIR group. The relative tumor volume of V/V_0_ and relative body weight of W/W_0_ were used to normalize the tumor volume and body weight change recorded every other day. Obviously, Figure 7E demonstrates that there was not a significant tumor growth inhibition effect in the control, MC@Lip, and US + NIR groups. By contrast, significant tumor growth inhibition was found in the MC@Lip + US group and MC@Lip + NIR group, indicating that MC@Lip-based SDT or PTT harbored a good anti-tumor effect. Notably, treatment with MC@Lip + US + NIR produced the strongest tumor growth inhibition performance, resulting in complete tumor regression in all tested tumor-bearing mice after 20 days (Figure 7D,E). Significantly longer survival time was also found in the MC@Lip + US + NIR group in comparison to other control groups, where all mice died within 18-26 days (Figure 7F). No obvious weight fluctuations could be observed for these mice in all groups over 20 days (Appendix A), showing the high therapeutic biosafety of the MC@Lip + US + NIR treatment. To evaluate the suppression effect of tumor metastasis, lungs in all groups were collected. Figure 7G–I clearly revealed that the MC@Lip + US + NIR group showed no noticeable metastatic tumors in the lungs, confirming that the MC@Lip + US + NIR treatment strategy substantially exhibited a robust performance against lung metastasis. Histological analysis of these tumors revealed that the apparent tumor destruction, the least Ki-67-stained cells, and most TUNEL-positive cells appeared in the tumors received with MC@Lip + US + NIR (Appendix A).

For the evaluation of the biosafety for MC@Lip, the in vivo toxicity of MC@Lip was detected by respective intravenous injection of MC@Lip or PBS into healthy Balb/c mice. No abnormal behavior change of those mice was observed in the 14-day period. Moreover, they were sacrificed for histological and blood examination on the 14th day. No histological morphology changes were recorded in the H&E staining sections of the main organs, including the heart, liver, lung, spleen, and kidney (Appendix A). In addition, it was found a negligible difference between MC@Lip- and PBS-administrated mice in the typical markers of liver and kidney function (such as ALT and BUN) and the red/white blood cells (Appendix A).

## 4. Discussion

Considering that tyrosinase is the crucial enzyme for the formation of melanin in the biosynthesis pathway, the tyr1 gene was synthesized and inserted into a heat-inducible gene circuit under the control of the pTlpA promoter. When the temperature is below 37 °C, the tyr1 gene would not be transcribed due to the repression of TlpA36 protein, which is constitutively expressed in the bacteria. Upon these bacteria being exposed to 37 °C, the TlpA36 repression protein would be deactivated and lose its binding with the pTlpA promoter, leading to the transcription and expression of the tyr1 gene. To test this, we transformed this plasmid with the heat-inducible gene circuit into the E. coli BL21 bacteria and incubated them at the different temperatures of 25 °C, 30 °C, or 37 °C. The SDS-PAGE result indicated that the tyr1 gene can express in a heat-inducible manner (Figure 2A). Next, we further examined the bacterium-mediated biosynthesis capability of melanin. The change of suspension color of different temperature-cultivated bacteria demonstrated that only the 37 °C cultivated bacteria could catalyze L-tyrosine into melanin. Interestingly, we also found the suitable particle size of biosynthetic melanin with 8nm under the 12 h incubation time with L-tyrosine. Additionally, the photo absorption and FTIR spectra of biosynthetic melanin nanoparticles were consistent with the natural melanin nanoparticles previously reported [34,35,36,37,38]. Thus, we successfully fabricated the melanin nanoparticles via the genetically engineered bacteria with the controllable particle size. Due to their relatively slower enzymatic reaction process than the chemical method for catalyzing the formation of melanin nanoparticles, it is especially favorable to obtain the melanin nanoparticles with suitable size by this biosynthetic way. We also found that the larger the melanin nanoparticles that were used to encapsulate into liposomes, the bigger the size of the MC@Lip would be, which was confirmed in Figure 3D. Given that the favorable small size is about 100 nm for tumor penetration via the EPR effect, we selected the 8 nm biosynthetic melanin nanoparticles to fabricate the MC@Lip. Together with the above results in Figure 3, we successfully fabricated MC@Lip with an appropriate particle size and high stability in physiological conditions, which were essential for the anti-cancer therapeutic application.

Since Ce6 and melanin in MC@Lip can function as the sonosensitizer and photothermal agents, respectively, we detected the sonodynamic and photothermal effects of MC@Lip. These data in Figure 4A–C indicated that ^1^O_2_ could efficiently be generated in the MC@Lip + US group due to the presence of Ce6. Moreover, we further investigated the photothermal effect of MC@Lip under NIR irradiation. We found that the ΔT in the MC@Lip + Laser group was higher than the free melanin and Lip group. Based on this result, liposomal confinement might effectively decrease the laser irradiation energy loss of free melanin in the solvent, which was in accordance with the previous study [38]. In addition, the suitable concentration and laser power density were optimized as follows: melanin concentration at 200 μg mL^−1^ and NIR laser power at 1.00 W cm^−2^ (808 nm). Generally, photothermal conversion efficiency (η) is considered as the gold standard for evaluating the photothermal capability of NPs for PTT. The η in this study was quantified to be 42.4%, which was in consistency to the reported melanin nanoparticles [30,34]. Collectively, these results provided strong support for MC@Lip to be applied as a promising sonosensitizer and photothermal agent with the efficient ^1^O_2_ production and photothermal conversion capability.

After the successful validation of the efficient ^1^O_2_ production and photothermal performance of MC@Lip, we subsequently examined their capability of PTT and SDT in vitro. The cytotoxicity of MC@Lip to 4T1 breast cancer cells and bEnd.3 normal cells were negligible, showing that MC@Lip was highly biocompatible. The fluorescence displayed in Figure 5B,C indicates the successful cellular internalization of MC@Lip and confirms the ROS generation of MC@Lip upon being received with US irradiation. Strikingly, the MC@Lip + US + NIR group showed the strongest cytotoxicity to 4T1 cells, with the lowest cell viability, which was further confirmed by the Calcein AM/PI staining assay.

Next, it has been well known that tumor imaging plays an essential role in image-guided therapy. These results in Figure 6 indicate that MC@Lip possesses good PA and fluorescence imaging performance, making it possible to visualize the aggregation of MC@Lip nanoparticles in the tumor. It is especially important for someone to choose the appropriate time to apply the US and NIR laser irradiation for treating the tumor.

Finally, we further examined the anti-tumor efficacy of MC@Lip-mediated PTT and SDT. This result in Figure 7B,C confirms that MC@Lip had the desirable PTT potential to ablate the tumor tissue. Collectively, the strongest tumor growth inhibition performance and longest survival time with no noticeable lung metastasis in the MC@Lip + US + NIR group confirmed that MC@Lip-based PTT-SDT harvested the therapeutical efficacy superior to that of monotherapy of PTT or SDT. No histological morphology and negligible difference in the typical markers of liver and kidney function further confirmed that MC@Lip had the favorable biosafety.

## 5. Conclusions

In summary, the melanin of the controllable particle size was successfully biosynthesized by the genetically engineered bacteria. The MC@Lip was then fabricated by using the biosynthetic melanin nanoparticles with appropriate size and Ce6. The resulting MC@Lip had approximately 100 nm particle size, with 74.71% ± 0.54% of encapsulation efficiencies for melanin and 94.52% ± 0.78% for Ce6 in MC@Lip. MC@Lip exhibited efficient ^1^O_2_ production and photothermal conversion capability upon receiving irradiation by US and NIR laser. When being irradiated by US or NIR laser, systemic administration of MC@Lip produced significant anti-tumor efficacy in both in vitro and in vivo experiments. Especially, US combined with NIR irradiation of tumors received with MC@Lip lead to complete tumor regression in all tested tumor-bearing mice after 20 days, indicating the great advantage of the combined use of SDT and PTT. More importantly, MC@Lip possessed good PA and fluorescence dual-modal imaging performance, making it possible to treat tumors under imaging guidance. Owing to the cellular origin of biosynthetic melanin, the better biocompatibility of MC@Lip will also help to pave the way to the future clinical translation.

## Figures and Tables

**Figure 1 pharmaceutics-15-02058-f001:**
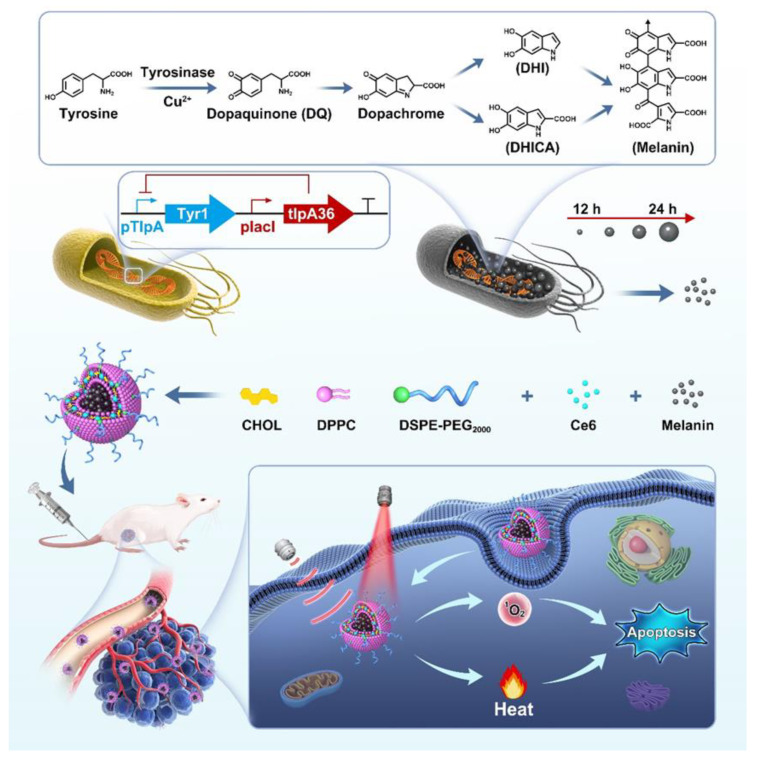
Schematic illustration of MC@Lip-based PTT-SDT. The melanin was synthesized by the bacteria genetically engineered with a temperature-inducible gene circuit. The rate-limiting enzyme Tyrosinase in the melanin biosynthesis pathway could be expressed at 37 °C and catalyze Tyrosine to produce melanin. The biosynthetic melanin and Ce6 were further encapsulated into liposomes to obtain MC@Lip. After being systemically administrated, MC@Lip could target the tumor site via the EPR effect. Owing to the presence of melanin and Ce6, NIR laser and US could be applied to the tumor, simultaneously producing PTT and SDT effects and greatly improving the anti-tumor efficacy. DHI: 5,6-dihydroxyindole, DHICA: 5,6-dihydroxyindole 2-carboxylic acid.

**Figure 2 pharmaceutics-15-02058-f002:**
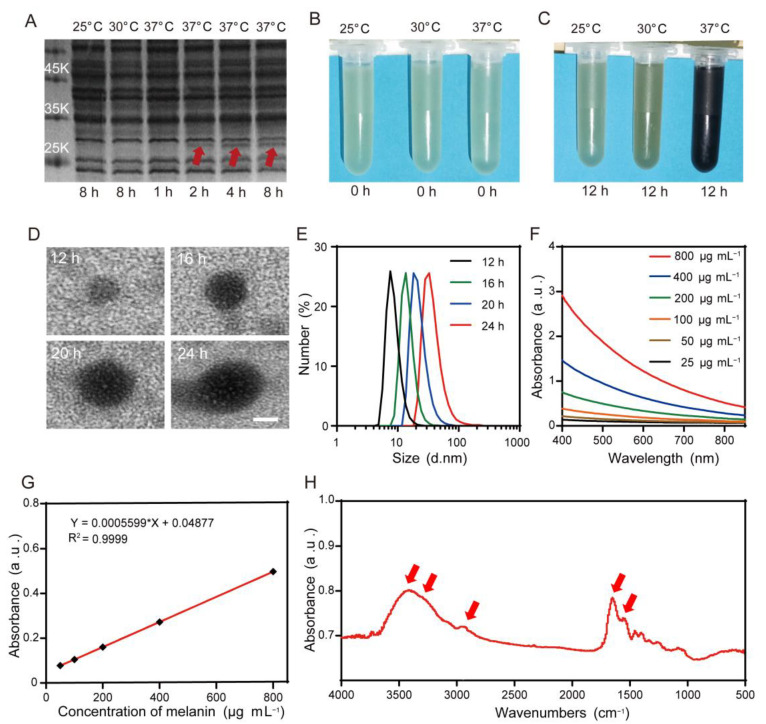
(**A**) The sodium dodecyl sulfate polyacrylamide gel electrophoresis (SDS-PAGE) revealed the expression of tyrosinase by the genetically engineered bacteria at the different temperatures of 25 °C, 30 °C, or 37 °C for various times. (**B**,**C**) The color changes of bacterial solution after adding L-tyrosine and CuSO_4_ into the genetically engineered bacteria incubated at the different temperatures of 25 °C, 30 °C, or 37 °C before (**B**) or after (**C**) 12 h. (**D**) Transmission electron micrograph showing the morphology of biosynthetic melanin, with size ranged from 8 to 35 nm. Scale bar = 10 nm. (**E**) Dynamic light-scattering analysis for the particle size of biosynthetic melanin. (**F**) The absorbance spectra in UV-Vis-NIR of biosynthetic melanin at different concentrations. (**G**) The linear relationship between the concentration of biosynthetic melanin and absorbance at the 808 nm wavelength. (**H**) FTIR spectra of biosynthetic melanin. Peaks (marked with red arrows) were 1546 cm^−1^ for C=C vibration in aromatic ring, 1683 cm^−1^ for C=O carbonyl group of carboxylic acid, 2946 cm^−1^ for poly O-H hydroxyl, 3300 cm^−1^ for N-H, and 3429 cm^−1^ for free O-H hydroxyl vibration.

**Figure 3 pharmaceutics-15-02058-f003:**
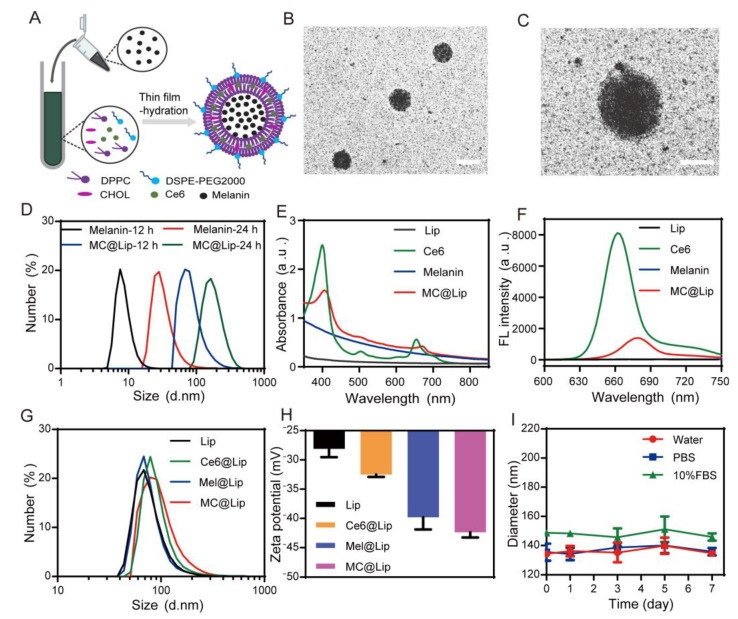
(**A**) Schematic illustration of the fabrication of MC@Lip. (**B**) TEM image of MC@Lip. Scale bar = 100 nm. (**C**) TEM image of single MC@Lip. Scale bar = 50 nm. (**D**) Dynamic light-scattering analysis of biosynthetic melanin and MC@Lip. (**E**) The absorption spectra in UV-vis-NIR and (**F**) the fluorescence spectra of liposome, Ce6, biosynthetic melanin, and MC@Lip. (**G**) Hydrodynamic sizes of Lip, Ce6@Lip, Mel@Lip, and MC@Lip. (**H**) Zeta potential of Lip, Ce6@Lip, Mel@Lip, and MC@Lip. (*n* = 3). (**I**) Changes in the diameter of MC@Lip stored in water, PBS, or 10% FBS at 4 °C for different durations.

**Figure 4 pharmaceutics-15-02058-f004:**
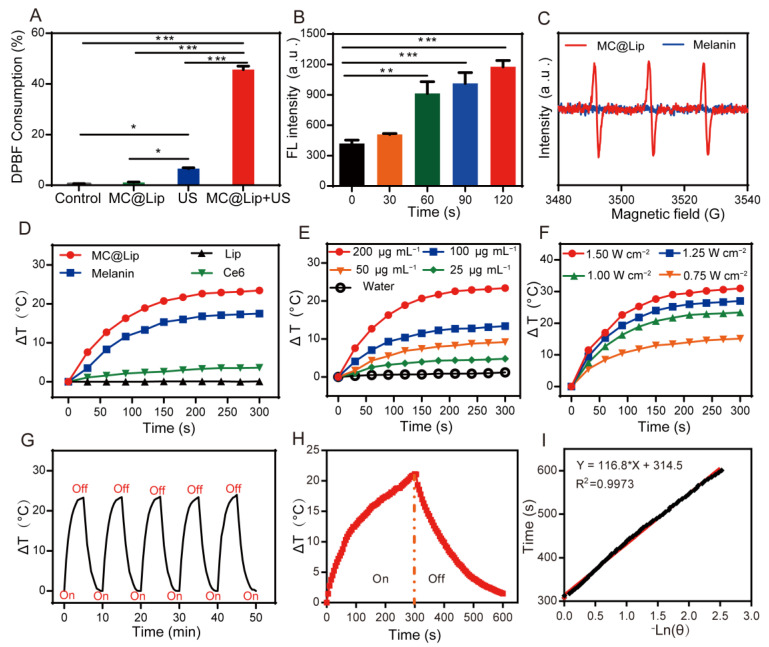
(**A**) Relative DBPF consumption of PBS or MC@Lip under US irradiation. (*n* = 3, * *p* < 0.05 and *** *p* < 0.001). (**B**) The ROS production of MC@Lip irradiated by US for different times. (*n* = 3, ** *p* < 0.01 and *** *p* < 0.001). (**C**) ESR signals of MC@Lip or biosynthetic melanin under US irradiation. (**D**) The plot of temperature elevation (ΔT) of MC@Lip (melanin concentration: 200 μg mL^−^^1^), Lip, melanin (200 μg mL^−^^1^), and Ce6 (12 μg mL^−^^1^) suspension exposed to the NIR irradiation (808 nm, 1.00 W cm^−^^2^, 5 min). (**E**) Plot of temperature elevation (ΔT) of pure water and MC@Lip suspension at a various concentrations (25, 50, 100, and 200 μg mL^−^^1^) under NIR irradiation (808 nm, 1.00 W cm^−^^2^, 5 min). (**F**) Plot of temperature elevation (ΔT) of MC@Lip (melanin concentration: 200 μg mL^−^^1^) suspension under various power densities of NIR irradiation (808 nm, 0.75, 1.00, 1.25 and 1.50 W cm^−^^2^, 5 min). (**G**) The plot of temperature elevation (ΔT) of MC@Lip (melanin concentration: 200 μg mL^−^^1^) irradiated by an 808 nm laser (1.00 W cm^−^^2^) for five irradiation cycles. (**H**) The plot of temperature elevation (ΔT) of MC@Lip (melanin concentration: 200 μg mL^−^^1^) that responded to the 808 nm laser (1.00 W cm^−^^2^) on/off. (**I**) Linear time data versus −Ln(θ) attained from the cooling stage after NIR laser off.

**Figure 5 pharmaceutics-15-02058-f005:**
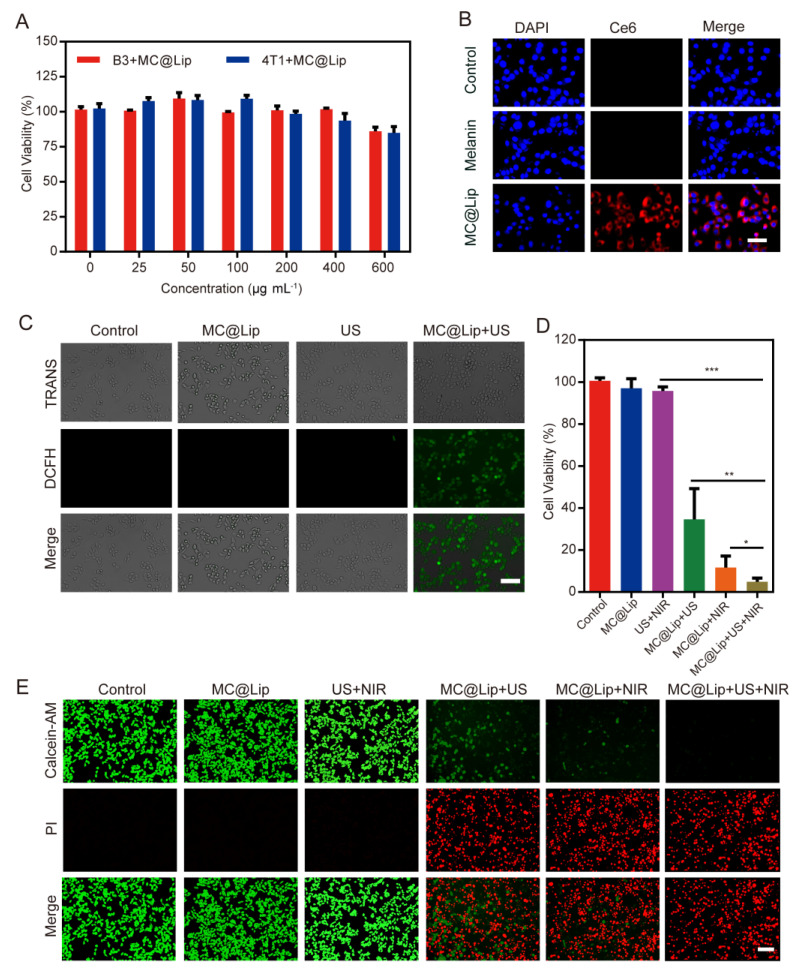
(**A**) Cytotoxicity of MC@Lip to bEnd.3 and 4T1 cells. (**B**) The fluorescence images of 4T1 cells stained by DAPI after the incubation with MC@Lip or melanin. Scale bar = 50 µm. (**C**) The fluorescence images of DCFH-DA-stained 4T1 cells after various treatments. Scale bar = 100 µm. (**D**) The viability of cells treated with or without MC@Lip under US (1.7 W cm^−^^2^, 1 min) and/or laser (1.00 W cm^−^^2^, 5 min). (**E**) The fluorescence images of cells under the above treatment stained with Calcein AM/PI assay. Scale bar = 100 µm. * *p* < 0.05, ** *p* < 0.01, *** *p* < 0.001.

**Figure 6 pharmaceutics-15-02058-f006:**
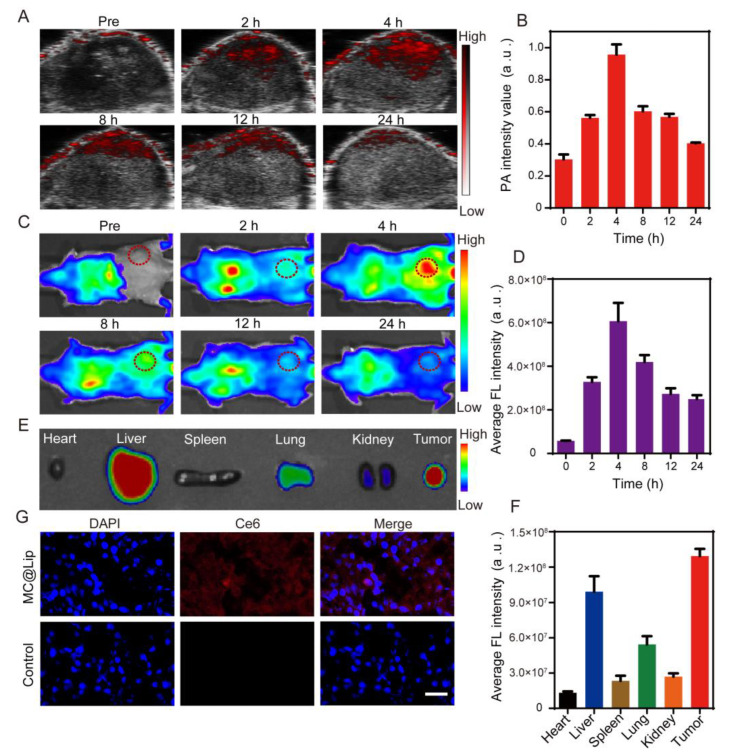
(**A**) PA images of tumors in the 4T1 tumor-bearing mice after the intravenous injection of MC@Lip at various time intervals. (**B**) The corresponding PA signal intensities of tumor regions in the 4T1 tumor-bearing mice. (**C**) The in vivo fluorescence imaging of the 4T1 tumor-bearing mice after the intravenous injection of MC@Lip at various time intervals. The location of the red dotted circle represented the tumor area. (**D**) The corresponding fluorescence intensities of tumor regions in the 4T1 tumor-bearing mice. (**E**) The ex vivo fluorescence imaging of the tumors and major organs at 24 h post injection. (**F**) The corresponding fluorescence intensities of the tumors and major organs at 24 h post injection. (**G**) Fluorescence microscope images of tumor sections at 24 h post injection of MC@Lip and the control group. Scale bar = 50 µm.

**Figure 7 pharmaceutics-15-02058-f007:**
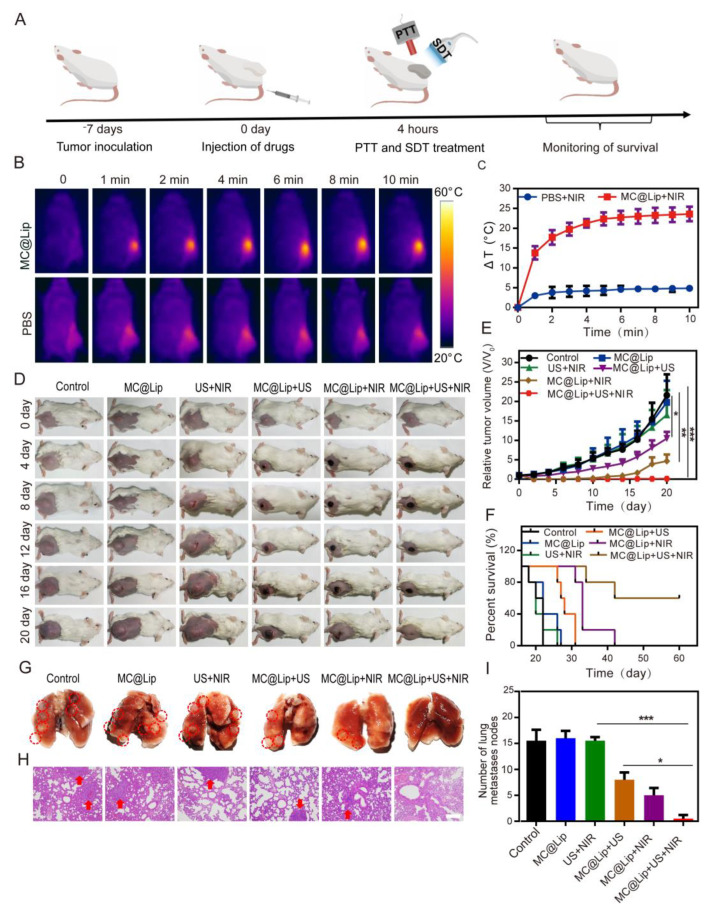
(**A**) The schematic illustration of in vivo experimental procedure of MC@Lip-mediated PTT and SDT. (**B**) The thermal images of mice treated with MC@Lip or saline under the NIR irradiation (808 nm, 1.00 W cm^−^^2^, 10 min). (**C**) The plot of temperature elevation (ΔT) of tumors received with MC@Lip or PBS under NIR irradiation (808 nm, 1.00 W cm^−^^2^, 10 min). (**D**) Photographs of the six groups of 4T1 tumor-bearing mice during a 20-day period after the different treatments. (**E**) Tumor growth curves (*n* = 5, mean ± SD, * *p* < 0.05, ** *p* < 0.01, and *** *p* < 0.001) of six groups after various treatments. (**F**) The percent survival of tumor-bearing mice after the various treatments (*n* = 5). (**G**) The representative photographs of tumor metastasis nodules in the lungs. The location of red dotted circle represented the tumor metastasis nodules in the lungs. (**H**) Representative H&E staining analysis of the lung metastasis nodules. The location of red arrows represented lung metastasis nodule. Scale bar = 200 μm. (**I**) The lung metastasis nodules numbers of six groups after various treatments (*n* = 3, mean ± SD, * *p* < 0.05 and *** *p* < 0.001).

## Data Availability

Not applicable.

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
