# Peer review of "Biosynthetic Melanin/Ce6-Based Photothermal and Sonodynamic Therapies Significantly Improved the Anti-Tumor Efficacy"

_pharmaceutics, 2023, doi:10.3390/pharmaceutics15082058_

Round 1
Reviewer 1 Report
This manuscript reports a novel combined strategy for tumor treatment using sonodynamic therapy (SDT) and photothermal therapy (PTT). The authors utilized genetically engineered bacteria to synthesize melanin. They encapsulated melanin and chlorin e6 (Ce6), a photosensitizer, into liposomes to create dual-functional SDT/PTT liposomes (MC@Lip). These liposomes showed efficient reactive oxygen species (ROS) production and photothermal conversion capabilities upon exposure to ultrasound (US) and near-infrared (NIR) laser irradiation. They demonstrated improved anti-tumor efficacy in vitro and in vivo. The data can clearly illustrate the design and the figures are nicely done. I believe this manuscript can be accepted by the Journal after minor revision. Please find the comments below.
1. In abstract, line 18, ultrasound should be stated rather than "US".
2. The authors need to provide a more explicit explanation for the in vivo experiment outlined in Figure 7A. The term "1/6 day" is unclear and needs to be elaborated upon. Additionally, it was mentioned that photodynamic therapy (PDT) was administered 4 hours post drug injection; however, no such information is available for sonodynamic therapy (SDT). Could the authors specify when SDT was conducted? Were PDT and SDT performed simultaneously? These points could use further clarification to ensure comprehensive understanding of the experiment.
3. In Figure 5C, the addition of results from the experiments involving NIR alone, MC@lip+NIR, and MC@lip+US+NIR would be beneficial in assessing the generation of reactive oxygen species (ROS). These extra experimental conditions will help provide a more comprehensive view of ROS production under different circumstances.
Minor editing of English language required
Author Response
Reviewer 1
- In abstract, line 18, ultrasound should be stated rather than "US".
Response: Thank you very much for the comments. According to the reviewer’s suggestion, ultrasound was stated for the abbreviation of "US" in the abstract (line 18-19 of Page 1).
- The authors need to provide a more explicit explanation for the in vivo experiment outlined in Figure 7A. The term "1/6 day" is unclear and needs to be elaborated upon. Additionally, it was mentioned that photodynamic therapy (PDT) was administered 4 hours post drug injection; however, no such information is available for sonodynamic therapy (SDT). Could the authors specify when SDT was conducted? Were PDT and SDT performed simultaneously? These points could use further clarification to ensure comprehensive understanding of the experiment.
Response: Thank you very much for the comments. According to the reviewer’s suggestion, we have provided a more explicit explanation for the in vivo experiment outlined in Figure 7A (line 494-499 of Page 14). In addition, the term "1/6 day" was used to illustrate the 4 h (4 hours=1/6 day) post-injection of MC@Lip in order to apply day for the same unit of time measurement. In Figure 6A and 6C, we had found that fluorescence imaging of Ce6 displayed the strongest fluorescence signal of tumor at 4 h post-injection, which was in consistent with the vivo PA imaging of MC@Lip. Thus, after 4 h post-injection of MC@Lip, the 4T1 tumor-bearing mice were conducted 5-minute US irradiation firstly and 10-minute NIR irradiation subsequently.
- In Figure 5C, the addition of results from the experiments involving NIR alone, MC@lip+NIR, and MC@lip+US+NIR would be beneficial in assessing the generation of reactive oxygen species (ROS). These extra experimental conditions will help provide a more comprehensive view of ROS production under different circumstances.
Response: Thank you very much for the suggections to improve our study. It takes the extra time to purchase reagents related to this experiment and the laboratory is relocated recently, we find it difficult to complete the additional experiment in a limited time for 10 days. Moreover, in this study, since Ce6 and melanin in MC@Lip can function as the sonosensitizer and photo-thermal agents, respectively, we detected the sonodynamic and photothermal effects of MC@Lip. According to the previous study, Ce6 as one of the current photosensitizers focused on the excitation wavelength in the visible light range (400-700 nm), which was usually at 660nm.
Sun R, Ma W, Ling M, Tang C, Zhong M, Dai J, Zhu M, Cai X, Li G, Xu Q, Tang L, Yu Z, Peng Z. pH-activated nanoplatform for visualized photodynamic and ferroptosis synergistic therapy of tumors. J Control Release. 2022 Oct;350:525-537. doi: 10.1016/j.jconrel.2022.08.050. Epub 2022 Sep 1. PMID: 36055597.
In this section, Ce6 was the significant sonosensitizer encapsulated in MC@Lip, which can generate ROS. Therefore, 2,7-Dichloro-fluorescein diacetate (DCFH-DA) as a ROS molecular probe was used to evaluate the intracellular ROS generation of MC@Lip under US stimulation. Similar experimental design can be found in the 30th reference of this study.
Zhang, J.;Shi, C.;Shan, F.;Shi, N.;Ye, W.;Zhuo, Y.;Zhang, Y.;Zhang, Z.;Shi, Y.; Peng, C. From biology to biology: Hematoporphyrin-melanin nanoconjugates with synergistic sonodynamic-photothermal effects on malignant tumors. Chemical Engineering Journal 2021, 408.

Reviewer 2 Report
After careful reading of the manuscript entitled “Biosynthetic melanin/Ce6-based photothermal and sonodynamic therapies significantly improved the anti-tumor efficacy” I have the following comments:
1- In the abstract, the abbreviation “PA” appeared without previous definition.
2- In the introduction section, line 68, please correct the following phrase “the limited tissue penetration of PTT usually results in complete ablation for some large tumors”.
3- Abbreviations included in Figure 1 should be define.
4- Please mention the full name of CCK-8 assay.
5- Section 2.3, please add the abbreviation of Fourier transform infrared spectroscopy next to it.
6- Please mention the source of Ce6. Is it synthesized by you or purchased?
7- In section 2.10, please mention the time at which you determine the cell viability after US+NIR+ MC@Lip application.
8- I suggest the authors to put the first paragraph in section 2.12, which explains the tumour induction, before section 2.11.
9- Section 2.12, Line 292, please correct “The tumor volume (V) = length × width2/2 was measured by the following formula: The changes of tumor volume and body weight”
10- Please mention in section 2.12 the time frame of your experiment.
11- Section 3.2, please correct “As the control groups, we also prepared four liposomal nanoparticles, including:” They are 3.
12- Section 3.2, authors reported “Like free melanin, Mel@Lip also exhibited the strong photo absorption from UV to NIR region” Results presented in Figure 3E are missing Mel@Lip.
13- Why there is DPBF consumption upon US application in Figure 4A?
14- Figure 4A and 4B please complete the statistical analyses by providing the analysis of each condition.
15- In Figure 4C, why you compared the MC@Lip with melanin for singlet oxygen production and not with the Ce6@Lip?
16- Why authors performed their experiment at 1 W for the NIR laser and not used higher laser power densities?
17- Section 3.3, please mention the value of the photothermal conversion efficiency.
18- Section 3.4, why the authors studied the uptake of melanin? Is it a free melanine or Mel@Lip?
19- Please correct the legend of Figure 5A, authors reported “Cytotoxicity of MC@Lip and melanin to bEnd.3 and 4T1 cells” No melanin results are provided.
20- Please improve the merge image of MC@Lip+US un Figure 5C.
21- Section 3.4, line 443, authors did not comment the results of MC@Lip+US+NIR of Figure 5D.
22- Could you please calculate the combination index to know if the treatments have synergistic or additive effects?
1- In the introduction section, line 73, please correct the English “Documents have also demonstrated that the combined use PTT with SDT”
2- Section 2.5, please correct the English “The measurement of hydrodynamic particle size distribution and zeta potential were used by a Malvern”, “The particle size of MC@Lip dispersed in phosphate …………….were recorded”
Author Response
Reviewer 2
- In the abstract, the abbreviation “PA” appeared without previous definition.
Response: Thank you very much for the comments. According to the reviewer’s suggestion, photoacoustic was stated for the abbreviation of "PA" in the abstract (line 29 of Page 1).
- In the introduction section, line 68, please correct the following phrase “the limited tissue penetration of PTT usually results in complete ablation for some large tumors”.
Response: Thank you very much for the comments. According to the reviewer’s suggestion, the phrase “the limited tissue penetration of PTT usually results in complete ablation for some large tumors” have been corrected by the sentence “the limited tissue penetration of PTT usually results in incomplete ablation for some large tumors” in the introduction section (line 69 of Page 2).
- Abbreviations included in Figure 1 should be define.
Response: Thank you very much for the comments. According to the reviewer’s suggestion, 5,6-dihydroxyindole and 5,6-dihydroxyindole 2-carboxylic acid have been respectively stated for the abbreviation of DHI and DHICA in figure 1 (line 112-113 of Page 3).
4- Please mention the full name of CCK-8 assay.
Response: Thank you very much for the comments. The full name of CCK-8 assay which was cell counting kit-8 have been added in the section 2.10 (line 119 of Page 4).
5- Section 2.3, please add the abbreviation of Fourier transform infrared spectroscopy next to it.
Response: Thank you very much for the comments. The abbreviation of Fourier transform infrared spectroscopy which was FTIR have been added next to it in the section 2.3 (line 153 of Page 4).
6- Please mention the source of Ce6. Is it synthesized by you or purchased?
Response: Thank you very much for the comments. Ce6 was purchased and the source was added in the section 2.1 (line121-122 of Page 4).
7- In section 2.10, please mention the time at which you determine the cell viability after US+NIR+ MC@Lip application.
Response: Thank you very much for the comments.The time to determine the cell viability after US+NIR+MC@Lip application was about 7-8 hours after the different treatments in the section 2.10 (line 254 of Page 6).
8- I suggest the authors to put the first paragraph in section 2.12, which explains the tumour induction, before section 2.11.
Response: Thank you very much for the comments. The first paragraph in section 2.12 had been inserted in the section 2.11 to explain the tumour induction (line 259-260 of Page 6).
9- Section 2.12, Line 292, please correct “The tumor volume (V) = length × width2/2 was measured by the following formula: The changes of tumor volume and body weight”
Response: Thank you very much for the comments. The sentense of “The tumor volume (V) = length × width2/2 was measured by the following formula: The changes of tumor volume and body weight” have been corrected by the sentense of “the tumor volume (V) was measured by the following formula: V = length × width2/2”. (line 295-296 of Page 7).
10- Please mention in section 2.12 the time frame of your experiment.
Response: Thank you very much for the comments. We have provided a more explicit explanation for the in vivo experiment outlined in Figure 7A (line 494-499 of Page 14). Additionally, the time frame of my experiment has been added to make it more definite. (line 299-300 of Page 7).
11- Section 3.2, please correct “As the control groups, we also prepared four liposomal nanoparticles, including:” They are 3.
Response: Thank you very much for the comments. “As the control groups, we also prepared four liposomal nanoparticles” was corrected by “As the control groups, we also prepared three liposomal nanoparticles” in the section 3.2 (line 356 of Page 9).
12- Section 3.2, authors reported “Like free melanin, Mel@Lip also exhibited the strong photo absorption from UV to NIR region” Results presented in Figure 3E are missing Mel@Lip.
Response: Thank you very much for the comments. “Like free melanin, Mel@Lip also exhibited the strong photo absorption from UV to NIR region” was corrected by “Like free melanin, MC@Lip also exhibited the strong photo absorption from UV to NIR region” in the section 3.2 (line 363 of Page 9). And “Compared to the Ce6@Lip or Mel@Lip, the MC@Lip remained a strong photo absorption of Mel@Lip and a new characteristic absorption centered at 403 nm corresponded well to that of Ce6” was corrected by “Compared to the Ce6 or melanin, the MC@Lip remained a strong photo absorption of melanin and a new characteristic absorption centered at 403 nm corresponded well to that of Ce6” in the section 3.2 (line 365 of Page 9).
13- Why there is DPBF consumption upon US application in Figure 4A?
Response: Thank you very much for the comments. Ultrasonic cavitation is a complex physical phenomenon, where the pressure changes induced by ultrasonic waves result in the formation of bubbles in the liquid. According to previous studies, under US irradiation, reactive oxygen species (ROS) can be generated due to cavitation effect, which can lead to DPBF consumption to some extent. Similar experimental design and output can be found in Figure 3A of following the 3th reference.
[1]Wang, M.;Choi, S.;Bai, Q.;Yu, K.;Guo, W.;Zhang, F.;Jia, R.;Qu, F.; Lin, H. Hollow Co-CeO2@PEG nanospheres: Ultrasound enhanced cascade-nanozyme for synergetic anticancer. Chemical Engineering Journal 2023, 456.
[2]Liang, S.;Deng, X.;Ma, P.;Cheng, Z.; Lin, J. Recent Advances in Nanomaterial-Assisted Combinational Sonodynamic Cancer Therapy. Adv Mater 2020, 32, e2003214.
[3]Huang, J.;Liu, F.;Han, X.;Zhang, L.;Hu, Z.;Jiang, Q.;Wang, Z.;Ran, H.;Wang, D.; Li, P. Nanosonosensitizers for Highly Efficient Sonodynamic Cancer Theranostics. Theranostics 2018, 8, 6178-6194.
14- Figure 4A and 4B please complete the statistical analyses by providing the analysis of each condition.
Response: Thank you very much for the comments. The statistical analyses by providing the analysis of each condition have been completed in Figure 4A and 4B.
15- In Figure 4C, why you compared the MC@Lip with melanin for singlet oxygen production and not with the Ce6@Lip?
Response: Thank you very much for the comments. The ESR analysis (Figure 4C) demonstrated the strong signal for TEMP solution containing MC@Lip after US irradiation, whereas no noticeable signal can be observed for TEMP solution of melanin with US. Therefore, the above result solidly confirmed Ce6 encasupted in MC@Lip can achieve the generation of 1O2 under US irradiation while melanin served as the photothermal agent to realize PTT.
16- Why authors performed their experiment at 1 W for the NIR laser and not used higher laser power densities?
Response: Thank you very much for the comments. The temperature increases of MC@Lip at various 808 nm laser power densities (0.75, 1.00, 1.25, 1.50 W cm-2) were also recorded with the fixed melanin concentration of 200 μg mL-1, showing the significant temperature elevation along with the increase of NIR irradiation power (Figure. 4F and S9). In addition, the suitable concentration and laser power density were optimized as follows: melanin concentration at 200 μg mL-1 and NIR laser power at 1.00 W cm-2 (808 nm) with high photothermal conversion efficiency (η) which was quantified to be 42.4%, while too high temperature can result in the excessive damage to the tissue surrounding the tumor.
17- Section 3.3, please mention the value of the photothermal conversion efficiency.
Response: Thank you very much for the comments. The value of the photothermal conversion efficiency had been mentioned in the discussion section (line 590-593 of Page 16).
18- Section 3.4, why the authors studied the uptake of melanin? Is it a free melanine or Mel@Lip?
Response: Thank you very much for the comments. It is a free melanin, and we found that the MC@Lip- but not melanin-incubated cells displayed the strong red fluorescence of Ce6 in cytoplasm, which confirmed the successful cellular internalization of MC@Lip through fluorescence detection.
19- Please correct the legend of Figure 5A, authors reported “Cytotoxicity of MC@Lip and melanin to bEnd.3 and 4T1 cells” No melanin results are provided.
Response: Thank you very much for the comments. The legend of Figure 5A which was “Cytotoxicity of MC@Lip and melanin to bEnd.3 and 4T1 cells” have been corrected (line 458 of Page 12).
20- Please improve the merge image of MC@Lip+US un Figure 5C.
Response: Thank you very much for the comments. The merge image of MC@Lip+US in Figure 5C have been improved.
21- Section 3.4, line 443, authors did not comment the results of MC@Lip+US+NIR of Figure 5D.
Response: Thank you very much for the comments. The results of MC@Lip+US+NIR of Figure 5D have been added in the Section 3.4 (line 453-454 of Page 12).
22- Could you please calculate the combination index to know if the treatments have synergistic or additive effects?
Response: Thank you very much for the comments. We found that the combined treatments had synergistic effects through calculating the tumor inhibitory rate of the MC@Lip + US group, MC@Lip + NIR group and MC@Lip + US + NIR group, which were 51.62%, 78.40% and 100%, respectively.
Comments on the Quality of English Language:
- In the introduction section, line 73, please correctthe English “Documents have also demonstrated that the combined use PTT with SDT”
Response: Thank you very much for the comments. The English “Documents have also demonstrated that the combined use PTT with SDT” had been corrected by he the English “Documents have also demonstrated that the combined use of PTT and SDT” (line 74 of Page 2).
2- Section 2.5, please correct the English “The measurement of hydrodynamic particle size distribution and zeta potential were used by a Malvern”, “The particle size of MC@Lip dispersed in phosphate …………….were recorded”
Response: Thank you very much for the comments. The English “The measurement of hydrodynamic particle size distribution and zeta potential were used by a Malvern”, “The particle size of MC@Lip dispersed in phosphate …………….were recorded” had been corrected by “The hydrodynamic particle size distribution and zeta potential were measured by using a Malvern Zetasizer Nano S90 instrument (Malvern, Worcestershire, UK). The particle sizes of MC@Lip dispersed in phosphate buffered saline (PBS), fetal bovine serum (FBS) or water were recorded after 0, 1, 3, 5 or 7 days.” (line 176-178 of Page 5).

Round 2
Reviewer 2 Report
I would like to thank the authors for their answer and efforts. I still have 2 comments:
1- Comment number 8 was not completely fulfilled the tumor induction procedure is still in section 2.12 (lines 276 to 280)
2- Figure 4A, the letter "A" is missing, further the statistical analysis between control and MC@lip+US is missing
Author Response
1- Comment number 8 was not completely fulfilled the tumor induction procedure is still in section 2.12 (lines 276 to 280)
Response: Thank you very much for the comments. The first paragraph in section 2.12 had been completely inserted in the section 2.11 to explain the tumour induction (line 260-266 of Page 6).
2- Figure 4A, the letter "A" is missing, further the statistical analysis between control and MC@lip+US is missing
Response: Thank you very much for the comments. The letter "A" of the figure has been added, and the statistical analysis between control and MC@lip+US group has been completed in Figure 4A.